# Vaccine Preventable Zoonotic Diseases: Challenges and Opportunities for Public Health Progress

**DOI:** 10.3390/vaccines10070993

**Published:** 2022-06-22

**Authors:** Ann Carpenter, Michelle A. Waltenburg, Aron Hall, James Kile, Marie Killerby, Barbara Knust, Maria Negron, Megin Nichols, Ryan M. Wallace, Casey Barton Behravesh, Jennifer H. McQuiston

**Affiliations:** Centers for Disease Control and Prevention, Atlanta, GA 30309, USA; pzy4@cdc.gov (A.C.); nvr6@cdc.gov (M.A.W.); esg3@cdc.gov (A.H.); gzk5@cdc.gov (J.K.); lxo9@cdc.gov (M.K.); bkk0@cdc.gov (B.K.); yfp3@cdc.gov (M.N.); gpg6@cdc.gov (M.N.); euk5@cdc.gov (R.M.W.); dlx9@cdc.gov (C.B.B.)

**Keywords:** vaccine, zoonotic, animal, emerging, One Health

## Abstract

Zoonotic diseases represent a heavy global burden, causing important economic losses, impacting animal health and production, and costing millions of human lives. The vaccination of animals and humans to prevent inter-species zoonotic disease transmission is an important intervention. However, efforts to develop and implement vaccine interventions to reduce zoonotic disease impacts are often limited to the veterinary and agricultural sectors and do not reflect the shared burden of disease. Multisectoral collaboration, including co-development opportunities for human and animal vaccines, expanding vaccine use to include animal reservoirs such as wildlife, and strategically using vaccines to interrupt complex transmission cycles is needed. Addressing zoonoses requires a multi-faceted One Health approach, wherein vaccinating people and animals plays a critical role.

## 1. Introduction

Scientists estimate that 60% of diseases in humans can be attributed to origins in and spreading from animals (i.e., zoonoses), and many of these diseases have high mortality rates and the potential to cause epidemics and pandemics [1]. Zoonotic diseases are responsible for approximately 2.7 million deaths and 2.5 billion human illnesses annually, in addition to impacting livestock production and food security [2]. SARS-CoV-2, which has zoonotic origins and emerged as a global pandemic in 2020, had caused over 4.4 M human deaths globally by mid-2021 alone [3]. In addition to human and animal health impacts, zoonotic disease burdens include substantial economic damage due to the cost of human illness, lost livestock production, and disrupted ecosystem health [4,5]. The potential for zoonotic diseases to affect human, animal, plant, and environmental health, global food security, and economic stability highlights the need for effective interventions that target prevention at multiple levels.

Vaccines are one of the most important public health achievements of the 20th century. From Edward Jenner using cowpox (*Variolae vaccinae*) to protect against smallpox in the 18th century, Pasteur’s discovery of how to inactivate the rabies virus to save human lives through vaccination, to the need to rapidly create effective vaccines during the explosive SARS-CoV-2 pandemic, vaccination is undeniably one of the most effective tools to prevent, control, and even eradicate disease [6,7]. Animal vaccines control diseases in companion animals, ensure safe food supplies through maintaining healthy livestock populations, and serve as an important barrier to prevent the transmission of some zoonotic diseases to humans [8,9,10,11]. Strategies for animal vaccination programs against zoonotic diseases have included vaccinating domestic animal species to prevent disease transmission to humans, and vaccinating wild animal species to prevent transmission to domestic animals and humans [12]. Developing new and improved vaccines to prevent the transmission of challenging or emerging zoonotic diseases represents an important future research horizon. 

For some zoonotic diseases, animal vaccines have been available for decades. When used as part of comprehensive prevention programs, these vaccines are highly cost-effective and could save human lives, improve animal health, and improve food and economic security [9,10,12]. Despite the successful historic use and current availability of animal vaccines, they are not always used effectively for maximum public health impact. For example, rabies vaccination campaigns for domestic animals and wildlife have nearly eliminated human rabies in developed countries [13]; however, globally, rabid dogs are still responsible for an estimated 59,000 human deaths annually, largely because many low-and middle-income countries lack the resources to develop comprehensive canine rabies vaccination programs [14]. 

Several limitations exist in the development and implementation of vaccination programs to protect against zoonotic diseases [12,15]. Challenges include defining the optimal population targets (e.g., animals vs. humans; domestic vs. wildlife animal reservoirs), navigating the complexities of vaccine licensing and approval for use (especially with different regulatory systems for human and animal products), and even the potential human risks of animal vaccine strains (e.g., human illness due to exposure to *Brucella* RB51 vaccine strains) [16,17].

In the United States, veterinary vaccines are regulated and licensed by the United States Department of Agriculture Center for Veterinary Biologics (USDA CVB). Following drug application, a product license is issued if all regulatory requirements are met after thorough evaluation during the application, production, and manufacturing processes. Following licensure, each new serial of a product is tested prior to marketing. The USDA CVB reviews the test results and has the option to conduct confirmatory testing prior to releasing the serial on the marketplace. 

Human vaccines are licensed by the US Food and Drug Administration (FDA) with immunization policies developed through the US Advisory Committee on Immunization Practices (ACIP). After licensure, the FDA continues to monitor vaccine production. Adverse events are voluntarily reported through the Vaccine Adverse Event Reporting System (VAERS), which was originally created in 1990 as a collaboration between the FDA and CDC to detect possible signals of adverse events associated with vaccines [18,19]. 

Vaccines intended for environmental use, particularly those that are modified or attenuated live vaccines, must ensure that all relevant sectors (e.g., human, animal, and environmental health agencies) agree on the safety and efficacy of the products and approaches for distribution. The CDC and USDA work closely to evaluate any potential risks associated with environmental distribution and to ensure appropriate scientific evidence is available for regulatory agencies to decide whether vaccine products can be used [20,21].

Pathogens and their vectors do not respect borders and can cause outbreaks of disease in wide geographic regions; however, the licensing of vaccines is country-specific. This complicates response activities and highlights the necessity of global collaboration to develop preparedness plans. The current regulatory landscape hinders information sharing and collaboration, slowing vaccine approvals and restricting availability.

Financial support for the development of safe and effective vaccines to prevent zoonotic diseases is an area that requires consideration and expansion by interested parties. Vaccine challenge initiatives present attractive opportunities to encourage the research and development of vaccines for zoonotic diseases. For example, the vaccine for use against *Brucella melitensis* in small ruminants, Rev-1, is an attenuated vaccine that can be pathogenic to humans and has accessibility challenges in low- to middle-income countries which are more affected by the disease [22]. The “Brucellosis Vaccine Prize” is a collaborative initiative to spur the development of an improved vaccine for *B. melitensis*, with increased safety and accessibility in low- and middle-income countries as the primary goal; the challenge is currently underway, with a potential grand prize of USD 20 million [23]. Information gained through research directed towards improving a vaccine for *B. melintensis* would also likely inform the improvement of the safety profile of a vaccine against *B. abortus*. Challenges including the “Brucellosis Vaccine Prize” can help define complicated needs around vaccines for zoonoses, encourage cross-sector collaboration, and drive the type of early research and development that can lead to innovative vaccine designs. Unlike veterinary vaccines, which can take as little as two years to reach the market (as was demonstrated by the West Nile Virus vaccine for horses), human vaccines can take 10–30 years from concept to licensure [24]. However, new vaccine platform technologies can help accelerate this process, as was the case with the development of vaccines for SARS-CoV-2. These platforms offer standardized, tested methods for delivering vaccine antigens, and can help shorten and streamline the development process. Assorted vaccine platforms, including viral vector technology and mRNA vaccines, were first used in veterinary medicine and have since been adapted for human vaccines [25]. 

Economic constraints and timing also pose challenges to animal vaccine development; evolutionary changes of the pathogen may render the vaccine ineffective, negating years or decades of research, or an outbreak may end before a newly developed vaccine comes to the market [26]. Outbreaks, such as those caused by the Rift Valley fever virus (RVFV), may be sporadic, and vaccination strategies must consider the disease ecology and be tailored to local circumstances whilst considering the risk, cost, and potential benefits. 

The concurrent development of animal and human vaccines against zoonotic diseases presents a unique opportunity for collaboration between historically separated biopharmaceutical industries [12,27]. However, due to differences in regulation and licensing, as well as avenues of funding, concurrent development may be challenging [12,28]. Increasing communication, collaboration, and information sharing between human and animal vaccine industries are important to help inform and drive new vaccine development. 

Vaccine-preventable zoonotic diseases present an opportunity to employ a One Health approach that involves a collaborative, multisectoral, and transdisciplinary approach—working at the local, regional, national, and global levels—with the goal of achieving optimal health outcomes recognizing the interconnections between people, animals, plants, and their shared environment [29]. A review of select vaccine-preventable zoonoses, current vaccine programs, and challenges for the development of vaccine-preventable zoonotic disease programs is provided. 

## 2. Review of Vaccine-Preventable Zoonoses

Given the high number of human diseases of animal origin and the public health’s strong reliance on vaccine programs to improve human health, it is no surprise that a listing of vaccine-preventable zoonotic diseases is extensive. These diseases have been studied for over a century, and the development of vaccines to control them represents remarkable achievements. In the United States, six of the top eight zoonotic diseases identified as those of greatest national concern have either human or animal vaccines, or both [30]. The top eight zoonotic diseases of greatest national concern in the US include zoonotic influenzas, anthrax, salmonellosis, West Nile virus, plague, severe acute respiratory syndrome coronavirus (SARS), rabies, and Rift Valley fever virus. Of the 30 One Health Zoonotic Disease Prioritization workshops conducted globally during the period 2014–2021, all five of the most commonly prioritized diseases (rabies, zoonotic influenza, brucellosis, Ebola and other viral hemorrhagic fevers, and anthrax) have either human or animal vaccines, or both [31]. However, in many cases, opportunities to more innovatively and effectively develop and use vaccines to address these and other zoonoses could be improved. 

### 2.1. Rabies

Perhaps no zoonotic disease represents the potential successes—and challenges—of preventing zoonotic disease transmission to humans such as rabies. Without appropriate vaccination, rabies is an invariably fatal, acute encephalitic disease caused by viruses of the Lyssavirus genus which is responsible for an estimated 59,000 human deaths annually [14]. Globally, the rabies virus variant that circulates in dogs presents the most serious public health threat among *Lyssaviruses* and is responsible for up to 99% of human deaths, mostly in children [14]. In the United States, human rabies cases declined precipitously following the implementation of canine vaccination regulations in the 1940s, and canine rabies was declared eliminated in 2007 [13]. However, the vaccination of dogs is still considered a high priority because *Rabies virus* variants are still present in several terrestrial meso-carnivore and bat species, and spillover from wildlife continues to pose an ongoing public health risk to humans, domestic dogs, and other animals in the United States.

Rabies management with oral rabies vaccination (ORV) has been successfully demonstrated with coyotes, gray fox, and raccoons, while canine rabies vaccination in the United States utilizes a parenteral (injection) approach. Ecological models and empirical evidence support that the vaccination coverage of at least 70% of free-roaming dog populations can eliminate the circulation of canine variants of rabies if maintained for a minimum of five years. A recent global analysis estimated that it would cost approximately USD 6.3 billion to eliminate canine rabies worldwide by 2030 through mass dog vaccination; this investment would prevent an estimated 500,000 human rabies deaths over the next 20 years [32]. However, meeting this ambitious goal would require a dramatic acceleration of funding and resource prioritization. 

The oral vaccination of dogs using live-attenuated viruses has gained attention as a new tool to advance dog-mediated rabies elimination [33]. This is a particularly attractive option for parts of the world where free-roaming dogs are challenging to catch and handle for parenteral vaccination. However, unfounded concerns regarding the safety and efficacy of well-studied ORV may delay large-scale implementation. The large-scale use of ORV for wildlife--which lacks motivation for use in dogs—presents a paradox for the rabies community. While further study and awareness campaigns to improve community acceptance may eventually enable the effective use of ORV for dogs, delays in the adoption of effective tools such as ORV enable the continued persistence of the rabies virus in dogs around the world and perpetuates otherwise preventable human deaths in many countries.

Despite the fact that vaccines for humans and animals have existed since 1885 and 1921, respectively, the rabies virus continues to cost an estimated USD 8.6 B USD per year globally [14,34,35]. Overall, 55% of the primary global economic burden of rabies is due to premature death, while 20% is attributed to post-exposure prophylaxis (PEP) use. Thus, preventing human deaths is a highly cost-effective strategy compared to the estimated cost of global elimination [14]. However, few countries in the past decade have made major advances in the elimination of the canine variant and the United States has not yet achieved the focal elimination of its most significant wildlife strain of rabies, namely the raccoon variant. Adequate funding, improvements in vaccine accessibility, and the inclusion of live-attenuated rabies viruses for oral vaccination are important tools that would help to see these goals achieved.

### 2.2. Brucellosis

Brucellosis is one the most common zoonotic diseases globally, impacting many species of animals, and is an example of both the opportunities and challenges of vaccine implementation. It also illustrates how in a contemporary vaccine prevention program, even highly successful efforts must evolve and modernize to address the changing disease landscape.

Historically, in the United States, *Brucella abortus* infections in cattle caused devastating chronic illness in people and significant economic losses in livestock due to abortions and fetal loss [36]. Animal vaccine programs were developed by the agriculture sectors to prevent agricultural losses and to protect human health by reducing the risk of zoonotic transmission. The US Brucellosis Eradication Program, established in 1934 and implemented by states through a complex set of laws and regulatory programs, exemplifies how successfully controlling the disease in cattle population has a direct impact on reducing the number of reported cases of human brucellosis [37]. 

Presently, brucellosis cases in humans are rarely reported in the United States and occur mainly in immigrants and travelers who are exposed to the disease outside of the United States [16,38]. *Brucella abortus* infection has been successfully eliminated from United States cattle herds as a result of this important program. The only remaining risk in the United States remains near Yellowstone National Park, associated with the possible spillover from bison and elk herds that were originally infected by cattle [39]. Additionally, vigilance is still required on the US–Mexico border to monitor for geographic expansion from that enzootic region. Despite the success of the eradication program and resulting negligible human risk of exposure throughout most of the United States, large numbers of cattle continue to receive modified live *Brucella abortus* vaccines in the United States today as directed by regulatory processes that were established to support the US Brucellosis Eradication Program [36].

Notably, an emerging risk factor for human brucellosis in the United States is the vaccine itself. Three live attenuated vaccines to prevent two *Brucella* species (*B. abortus* and *B. melitensis)* in animals are commercially available [40,41]. While less pathogenic than wild-type *Brucella* species, all three vaccines are pathogenic in humans and can cause clinical illness. Currently, the *B. abortus* vaccine RB51 is actively used throughout the United States, and it was considered less pathogenic (and therefore safer to humans) than the prior historic vaccine Strain 19 [42]. Over time, the use of the *Brucella* vaccine, as well as the rapid identification of animals and removal from affected herds, contributed to the United States’ highly successful elimination of *Brucella* from US cattle herds [36]. With its widespread use, however, accidental exposures to the vaccine, such as veterinarian exposures to RB51 through needlestick injury, continue to occur. [43,44,45]. 

An emerging public health risk of concern from RB51 has been identified and is associated with a rise in popularity among the general public of consuming unpasteurized (raw) milk and milk products within the United States. This has resulted, in part, from changes in laws permitting raw milk sales in retail stores or directly to consumers in over half of US states [46]. As a result, RB51 cases have been recently diagnosed in people who consume unpasteurized dairy products obtained from previously vaccinated animals [16,17,47]. In this setting, some small percentage of cows that are vaccinated in calfhood as part of state and federal programs never clear the vaccine strain and develop a chronic infection, shedding infectious bacteria in milk during later lactation stages [36,48]. Several recent human cases of RB51 linked to raw milk have been reported, and in the United States, the general public is at higher risk of RB51 than wild-type infection, especially when consumer preferences remove protective pasteurization processes from milk and dairy products [16,17]. Currently, the legislation related raw milk is enacted at a state level, and interstate sales are illegal [46]. For recent cases of RB51 among humans in the United States, both implicated cows were Jersey breeds, and the chronic shedding of RB51 post-vaccination is believed to be more common in some individual animals of Jersey breed [17,47].

An initial risk assessment of the RB51 vaccine was performed to evaluate the impact of the vaccine on the safety of animals, public health, and the environment [49]. The USDA Animal and Plant Health Inspection Service (APHIS) also conducted an environmental assessment and concluded that the licensure of RB51 would not significantly affect the quality of the human environment [49]. However, the safety assessment was conducted during the time when farmers and veterinarians were populations at the greatest risk of exposure to RB51 from accidental vaccine sticks, and before the rise in popularity of consuming raw milk and milk products among the general public in the United States. The assessment was also conducted to ascertain safety compared to the older *Brucella* vaccine Strain 19, which was more highly pathogenic to humans and livestock. The changes in the environmental risk of exposure to *Brucella* in the United States, as well as the changing demographics of at-risk populations, represent an expanding gap that has developed since the original vaccine assessments were performed.

RB51 was developed as a vaccine that could differentiate infected from vaccinated animals (DIVA), an important tool for surveillance purposes. However, because traditional serologic methods to detect brucellosis were not designed to identify RB51 in either humans or animals, and symptoms can be non-specific and wax and wane, human cases are undoubtedly underdiagnosed and underreported [44,50]. Additionally, RB51 is resistant to rifampin and penicillin, which are the first-line medications used to treat human brucellosis, further complicating human treatment and diagnosis [44,51]. While the CDC established a passive surveillance system from 1998 to 1999 to identify cases of accidental RB51 exposure among veterinarians and farmers, this system did not evaluate possible RB51 infection from exposure to unpasteurized dairy products [45]. National surveillance for human *Brucella* infections is tracked through the CDC’s National Notifiable Diseases Surveillance System (NNDSS), and brucellosis cases are commonly investigated by state health authorities to assess possible sources of domestic or international exposure [52]. However, because it is so challenging to detect human infections with RB51, which may be milder than traditional cases of brucellosis, it is uncertain whether current surveillance practices are sufficient for detection and reporting through the NNDSS, and unclear how widespread the human RB51 infection caused by raw milk consumption may be. This gap may become increasingly problematic as consumer preferences continue to evolve [45,53]. Further developments to enhance vaccine safety to produce non-pathogenic animal vaccine strains should be considered an important future research horizon if the consumption of unpasteurized dairy products becomes more widespread in the United States. Additionally, changes in vaccine program administration in the context of emerging human health risks should be discussed collaboratively between the public health and agriculture sectors. For example, implementing screening programs on raw milk dairies and antigen-based tests could provide targeted testing to help protect public health. Similarly, the current state requirements for *Brucella* vaccination programs could be re-examined, particularly for the cows used for raw milk dairies and in breeds at risk for chronic shedding. 

### 2.3. Coronaviruses

Nothing illustrates the potential perils of zoonotic disease risks as dramatically as the COVID-19 pandemic. Coronaviruses include several known zoonotic pathogens, most notably the Middle East respiratory syndrome coronavirus (MERS-CoV) and severe acute respiratory syndrome coronavirus 1 and 2 (SARS-CoV-1 and SARS-CoV-2). These viral pathogens highlight the challenges and opportunities for vaccine use to prevent emerging zoonotic transmission, and the necessity of taking a flexible and adaptive approach. MERS-CoV is a relatively rare zoonotic infection with epidemic potential that has caused severe respiratory illness in people. Multiple lines of evidence point to dromedary camels as a reservoir host [54,55,56,57]. Three potential vaccination strategies to prevent MERS-CoV infection are currently under consideration: (1) the vaccination of camels to prevent the camel-to-human transmission of MERS-CoV; (2) the vaccination of specific at-risk human populations (e.g., healthcare personnel and those with occupational exposure to camels); and (3) the reactive use of a vaccine in at-risk humans as a means of outbreak control. Camel vaccination would likely provide minimal benefit to the animals [58], but given the severity of disease in humans, the lack of viable treatment options, and epidemic potential, camel vaccination is still considered a viable preventive measure for human infection. Vaccine trials for MERS-CoV are underway in both camels [59,60] and humans [61,62], with one candidate, the ChAdOx1 MERS CoV vaccine, undergoing trials across both camels and humans [59,62].

In contrast, SARS-CoV-2, the virus that causes COVID-19, while suspected to have originated in an animal reservoir, has reached pandemic levels through sustained person-to-person transmission. Additionally, transmission from humans to a wide range of mammalian animals has been documented, including companion animals (dogs, cats, and ferrets), exotic animals (lions, tigers, and other large cats, non-human primates, otters, and multiple other species), production animals such as farmed mink, and most recently, free-ranging white-tailed deer, and additional susceptible animal species continue to be identified [63,64]. At the time of publishing, the Pfizer-BioNTech COVID-19 Vaccine and Moderna Spikevax vaccines have been approved by the FDA for use in humans, and the Johnson & Johnson has been approved under emergency use authorization (EUA) in the United States [65,66]. An experimental animal vaccine developed by Zoetis has been authorized for use in mink herds and zoo animals that have demonstrated natural susceptibility to SARS-CoV-2 infection [67]. Given the potential for the zoonotic transmission of SARS-CoV-2 from mink to humans, in addition to the emergence of novel viral variants in this animal species documented in Europe [68], vaccinating mink as a mechanism to protect both animal and human health has been implemented in the United States. This initiative has, in part, been fueled by concerns about vaccine and therapeutic efficacy in the context of mink-associated viral variants which resulted in the mass culling of the mink population in Denmark [68].

While the vaccination of people primarily serves to protect them against illness and death, it has the indirect benefit of potentially preventing zoonotic transmission from people to animals. Prior reports have documented the transmission of SARS-CoV-2 from humans to critically endangered and valuable zoo animal species [69]. Experimental animal vaccines have been used to protect these species from disease transmission from humans [70]. The finding of widespread SARS-CoV-2 transmission among white-tailed deer in the United States and Canada following the likely spread from people has raised concerns about whether the virus might become enzootic in this species, which could lead to further viral evolution and the emergence of new mutations or variants with public health impacts [63,71]. Further research is needed to better elucidate any indirect protection that vaccinating people may provide to animals. Preventing inter-species transmission from people to animals and among animals, which may in part be accomplished by vaccinating humans, could reduce the emergence of variants and help preserve vaccine efficacy. SARS-CoV-2 demonstrates the complexities of inter-species disease transmission, further highlighting how prevention and control efforts such as vaccination should reflect the shared threat of disease among humans and a variety of animal species.

### 2.4. Influenza 

For influenza viruses, human and animal vaccines play critical roles in protecting the individual, preventing zoonotic transmission, and preventing virus reassortment (i.e., recombining larger gene segments from different viruses) that may lead to a novel virus with pandemic potential. In addition to humans, influenza A viruses are known to routinely circulate in six animal species or groups (i.e., wild water birds, domestic poultry, swine, horses, dogs, and bats), and can infect many other animal species through inter-species transmission [1,2]. 

Influenza A viruses are constantly evolving, making it possible on rare occasions for animal influenza viruses to change in such a way that they can easily infect people [3,4]. As with SARS-CoV-2, influenza vaccines not only directly protect the recipient, but also prevent inter-species transmission, which can facilitate the emergence of new viruses [1,5].

Many influenza A viruses are not host specific; for instance, humans and some animals, such as pigs and dogs, can be infected with either swine, human, or avian influenza viruses. If a host is infected with different viruses at the same time, it is possible for reassortment to occur and create a novel virus. While it is unusual, sporadic human infections caused by certain avian influenza viruses and swine influenza viruses have been reported. In combination with other measures, vaccines are a valuable tool in preventing the zoonotic transmission of influenza viruses. 

In the United States, in addition to seasonal influenza vaccines for humans, there are 12 animal influenza vaccines licensed through the United States Department of Agriculture’s Center for Veterinary Biologics (USDA CVB). Vaccines are available for domestic poultry (3), swine (5), horses (1), and dogs (3), in function of the needs of the individual animal owners, whether they be commercial animal production facilities, family agriculture farms, or companion animal owners [6]. Swine are vaccinated, while birds are not, and the lack of an oral vaccine for the mass vaccination of poultry presents a challenge for the poultry industry. To be most effective, it is important for the animal influenza vaccines to antigenically match the influenza virus strain currently circulating in the respective animal species.

A candidate vaccine virus (CVV) is an influenza virus that has been prepared by the Centers for Disease Control and Prevention (CDC)—or another public health agency—that can be used by vaccine manufacturers to produce an influenza vaccine for humans [7]. In addition to preparing CVVs for human seasonal influenza vaccine production, the CDC routinely develops CVVs for novel avian and swine influenza viruses with pandemic potential as part of their pandemic preparedness activities [8]. Influenza vaccines protect against specific influenza viruses, with minimal cross protection against other influenza viruses, so the first step in creating a CVV against a particular avian or swine influenza virus is to identify the emerging animal influenza virus that is posing or may pose a risk to human health [8,9,10]. 

The tendency of influenza A viruses for mutation and reassortment, combined with the presence of wildlife reservoirs, the circulation of avian influenza viruses in domestic poultry, and the diversity of swine influenza viruses in the domestic swine population, make vaccine development a challenge. However, influenza vaccination in animals remains an important measure to control and prevent reassortment events and zoonotic transmission, and to reduce influenza illness in humans and animals.

## 3. Implementing Strong Global Programs for Vaccine Preventable Zoonoses

Efforts to prevent and address zoonotic diseases must reflect their global nature and use a One Health approach because with worldwide trade, travel, and the movement of animals and animal products, zoonotic pathogens anywhere are a threat to animal and human health everywhere. The World Health Organization (WHO) Initiative to End Dog-Mediated Human Rabies Deaths by 2030, a collaboration led by the WHO, World Organisation for Animal Health (OIE), and the Food and Agriculture Organization (FAO), is one of the most prominent collaborative global efforts to employ vaccines to eliminate a zoonotic pathogen as a public health threat [32,72]. As a One Health effort, this strategic plan heavily relies on dog vaccination and human PEP to prevent transmission and disease. This initiative provides structure and impetus for countries to improve their rabies control and elimination programs and has spurred additional efforts, such as the Global Alliance Vaccine Initiative (Gavi), to help achieve this goal [73]. 

Gavi has committed itself to expanding access to human rabies vaccines for PEP to provide equitable access to persons following a suspected dog bite [73]. Together, these programs are examples of comprehensive One Health approaches using veterinary and human vaccines to prevent zoonotic disease: veterinary vaccines used to prevent disease and transmission between dogs and from dogs to humans, and PEP used to prevent disease in humans in the event of exposure. Focusing solely on human PEP fails to address the complete epidemiologic picture and will result in the need for never-ending, costly PEP provision. The vaccination of dogs without the expanded provision of PEP to bite victims will eventually eliminate the threat to humans, but would likely result in hundreds of thousands of human deaths as canine vaccination programs are implemented [74]. Oftentimes, zoonotic disease control efforts are siloed, resulting in unequal control capacities and disease persistence; Gavi is setting a clear example of the importance of recognizing a One Health approach to address rabies control from both a human and animal perspective.

Within the United States, the CDC Global Immunization Strategic Framework (GISF) provides guidance for CDC’s work during the next ten years to advance the control, elimination, and eradication of vaccine-preventable diseases [75]. For the first time, vaccine-preventable zoonoses are identified as a priority area within this framework. The GISF identifies opportunities for collaboration, the remaining challenges, and ways to maximize the efficiency and effectiveness of disease prevention interventions. The GISF emphasizes the importance of taking a comprehensive, One Health approach that is flexible, adaptable, and sustainable. Recognizing the importance of a collaborative multi-sectoral approach and the potential of using veterinary vaccines to prevent zoonotic disease is a foundational step in implementing these plans.

The global threat of vaccine-preventable zoonoses prevention programs is demonstrated by Rift Valley fever virus (RVFV). RVFV is transmitted by the *Aedes* species mosquito, and competent vectors exist far outside of its current distribution, which is primarily limited to Africa and the Arabian Peninsula [12]. The introduction and establishment of novel mosquito-borne diseases in the United States remain a risk, as historically demonstrated by the introduction of the West Nile virus into the United States in 1999, followed by its rapid spread across the country [76,77]. The USDA considers RVF to be of the most significant foreign animal diseases posing a threat to the United States due to the presence of compatible *Aedes* vectors and possible economic impacts on the livestock industry [78,79]. During RVFV outbreaks, large numbers of infected livestock serve as an amplifier for viral spread. Humans are primarily exposed to RVFV through contact with blood or other body fluids of infected livestock. Thus, vaccinating livestock is central in stopping the transmission cycle of RVFV by simultaneously preventing further transmission among mosquito vectors, in wild animals, and preventing additional human infections [12]. Several animal vaccines currently exist, but their use is complicated by the sporadic nature of outbreaks, which is often related to weather events such as flooding. Geographic locations prone to RVFV outbreaks are frequently rural and economically challenged, and so thus also present challenges with regard to the vaccine cost, availability, and licensure, which may not extend to the entire geographic area experiencing disease. Ideally, vaccines should be DIVA to allow animal exports to resume following outbreaks and to better track the distribution of the natural disease. The tremendous potential for RVFV to cause disease across broad geographic regions or be imported to new areas, coupled with the lack of a global plan to address this disease, highlights the necessity of developing improved vaccine strategies that are tailored to local risk and reflective of disease ecology.

## 4. Co-Development of Human and Animal Vaccines

An example of vaccine co-development, or the concurrent development of a vaccine intended for veterinary and human use against a common pathogen, occurred in 1999 with the West Nile virus (WNV) vaccine [80,81]. The West Nile virus was first detected in animals in 1999, and quickly spread across the United States, causing illness in humans, horses, and numerous bird species [76]. During the early phase of the epidemic, veterinary and human vaccine candidates were developed in tandem, and the data gathered from both programs were shared across sectors to help inform the pathway to advanced clinical trials [12]. In two years, a veterinary vaccine was developed for use in horses, and one of the developed vaccines was able to protect California Condors from extinction [12,82]. In this instance, the development of the veterinary vaccine helped inform the development of the human vaccine by providing information on the safety, duration of immunity, and immune protection [12]. The cost of developing an animal vaccine is approximately 10% of that of developing a human vaccine [83]; the WNV vaccine co-development provides an example of the potential for collaborative vaccine development through coordination between human and veterinary sectors. Today, while the WNV vaccine is considered a core vaccine in horses, there is currently no licensed vaccine for humans due to obstacles with obtaining efficacy data because of unpredictable WNV transmission patterns. Approval of a human WNV vaccine would likely require the use of the FDA’s Animal Rule, which provides a pathway to licensure for vaccines using well-designed animal studies when human clinical trials are not possible or are unethical [84]. In addition to obstacles to conducting Phase III studies, initial estimates of a universal vaccination program among humans in the United States was not cost-effective. However, more recent cost-effectiveness analysis in the United States has demonstrated that the implementation of a human WNV vaccine program that targets persons aged ≥60 years (who are at increased risk for severe outcomes) and who live in high-incidence areas would improve the cost-effectiveness of the program and could reduce human deaths by 60% [85,86,87,88]. 

Another example of vaccine co-development is LYMErix, a vaccine that was developed in the 1990s for human application to protect against Lyme disease. LYMErix was based on a novel approach that targeted an outer surface protein, OspA, to inactivate *B. burgdorferi* while still in the tick vector. Despite a reported 76% efficacy in the first year, LYMErix was removed from the market after four years, with the manufacturer citing poor sales performance, amid unsubstantiated concerns of side effects and an ongoing class-action suit [89]. Meanwhile, several OspA vaccines were developed and are currently available to protect against *Borrelia burgdorferi* infection in dogs [90]. A second generation OspA vaccine candidate for human use, Val15, is undergoing stage 2 clinical trials [91]. OspA vaccines have also been evaluated for possible use in wildlife reservoirs of *B. burgdorferi*; however, there remain significant barriers to vaccine delivery. The similarities between these vaccines highlight the potential for co-development opportunities through sharing technology that has been in use for years in animal applications. As Lyme disease causes a substantial burden of illness among humans with approximately 476,000 Americans being diagnosed and treated annually, vaccine development remains an important strategy for the prevention and control of this zoonotic disease [92,93]. 

An important example of zoonotic disease vaccine co-development with wide-ranging benefits was the development of the ChAdOx1 MERS-CoV vaccine. This vaccine, an adenovirus vector vaccine with the MERS-CoV spike protein was developed for use in both humans and camels. Immunogenicity and efficacy results were first reported in camels in 2019 [59], and phase 1a results were reported in humans by Folgatti et al. in early 2020 [62]. Additionally, in early 2020, scientists from the same group reported the development of an adenovirus vector vaccine with the spike protein of SARS-CoV-2; the ChAdOx1 nCoV-19 vaccine, AZD1222 [94]. The work on a functional MERS-CoV vaccine therefore enabled rapid scientific translation into a vaccine for a novel emerging coronavirus. The AZD1222 vaccine has since been used globally, with over 2 billion doses distributed in less than 12 months after initial approval [95]. The use of the AZD1222 vaccine in turn, has informed the further development of the ChAdOx1 MERS-CoV vaccine, with the recently published human phase 1b results [96]. Additionally, it is notable that the rapid early development of mRNA vaccines for SARS-CoV-2 was also enabled by the previous development of vaccines for MERS-CoV [97]. More specifically, structural studies using MERS-CoV led to the development of mutations that stabilized beta coronavirus spike proteins in the prefusion state, improving their expression, and increasing immunogenicity [98]. This principle was then applied to design mRNA-1273, an mRNA vaccine that encodes a SARS-CoV-2 spike protein that is stabilized in prefusion conformation [97].

In unique situations where the goal is the development of a human drug or biologic product, and where it would be unethical to deliberately expose healthy human volunteers to a lethal or permanently disabling toxic biological, chemical, radiological, or nuclear substance, the FDA “Animal Rule” is intended to expedite the development of new drugs and biologic products as medical countermeasures (MCMs) [82]. This rule permits the FDA to approve drugs or license biological products on the basis of animal efficacy studies for use in ameliorating or preventing serious or life-threatening conditions caused by exposure to lethal or permanently disabling toxic substances [84]. Additionally, the use of animal models for the pre-clinical testing of human vaccines provides another example of an opportunity for information sharing across sectors. Thus, even when vaccine co-development is not the intention, collaboration and information sharing across the animal and human sectors can inform the development of novel drugs and biologics. 

## 5. Discussion

The multi-factorial nature of zoonotic diseases highlights the need for One Health collaborations to protect human, animal, and environmental health. Despite the availability and proven efficacy of using vaccines to prevent zoonotic diseases, they are not always used to their full potential. Although opportunities for partnership exist regarding intervention strategies, vaccine target identification, and vaccine development, they are underutilized. To aid in this effort, it would be beneficial if information regarding vaccine research and development were shared between human and animal health sectors. This might be accomplished through opportunities for interprofessional education at conferences and events, and include information on how recent developments in vaccine science can impact multiple species and promote One Health collaborations for examining vaccine efficacy.

Historically, human and animal health sectors have primarily focused on preventing diseases that affect their respective species. Consequently, prevention plans are often limited in scope and reactive in nature, resulting in the continued burden of zoonotic diseases. Proactive collaboration between human and animal sectors such as having agreements for sharing biological samples and laboratory results prior to disease emergence or outbreaks would expedite the sharing of DNA sequences and information to aid in more timely prevention strategies. This proactive approach by federal, state and local animal and public health entities may help to achieve more timely disease control.

Human and animal health agencies have their own specific knowledge and tools, including surveillance data, which are needed to direct efforts to prevent these diseases in their respective species, but the collaborative One Health approach needed to address zoonotic diseases is lacking. Although rabies is one of the oldest vaccine-preventable zoonotic diseases, and arguably has employed the most expansive coordinated approach among vaccine preventable zoonoses, this virus remains a significant human health threat across the world. Despite the strong achievements of the past century, including the development and wide availability of efficacious canine and human vaccines, global control has remained elusive, with tens of thousands of deaths annually. The new and energized focus on rabies prevention by the WHO, along with Gavi and other partners, represents a change in the approach to rabies control, particularly because the architecture of the plan involves both human and animal vaccine programs and sets ambitious global goals requiring coordinated work across human and animal health sectors. This type of core coordination would also strongly benefit other vaccine-preventable zoonoses. 

Currently, human and animal health divisions within the biopharmaceutical industry are kept largely separate, even with regard to zoonotic diseases. As exemplified by WNV, the co-development of human and veterinary vaccines can provide an opportunity for collaboration and information sharing, including safety and efficacy studies, the duration of immunity, and immune protection. Ideally, research, funding, and communication efforts should reflect the shared nature of zoonotic diseases for which vaccines are being developed. As both sectors bear the burden of zoonotic diseases, directly or indirectly, it follows that they should share the responsibility and cost of developing and implementing prevention and mitigation strategies. Additionally, in developing vaccines and communicating about their importance to disease prevention, developing plans for unified communication and messaging about vaccine technology (e.g., mRNA) may help jointly combat misinformation. Sharing information regarding the use of vaccines in veterinary patients might help expand public acceptance for human vaccines, during a time of increasing vaccine hesitancy in some individuals [99]. Sector-specific approaches to zoonotic disease prevention and control efforts that do not consider the complex landscape of disease ecology and multi-host diseases using a One Health approach miss opportunities for more significant success. 

While a comprehensive sector-inclusive approach to utilize vaccines to prevent zoonotic diseases is undoubtably more complicated than solely focusing on human or animal populations, the careful examination of disease reservoirs, vector ecology, and transmission dynamics may identify unique opportunities for effective intervention. Examples of past successful vaccine development and implementation, such as the elimination of canine-mediated rabies in the United States, and the use of *Salmonella* Enteritidis vaccination among egg-laying poultry flocks demonstrate the promise of using vaccines to prevent zoonotic diseases. Veterinarians are uniquely suited to this task and can aid in the adoption of a more inclusive approach to vaccine utilization by working with producers to examine the potential implications for public health when considering whether to utilize certain vaccines in a flock or herd of animals.

The diseases described herein present unique challenges that require comprehensive plans to effectively reduce zoonotic disease impacts. While there is a great deal of work happening for some pathogens, such as rabies, for others, there is lack of forward momentum or a generalized acceptance of background disease rates as “something to live with”. The diseases listed in Appendix A represent partially developed programs. Each also represents opportunities for the renewed consideration of more comprehensive and collaborative approaches for prevention and control. Vaccination as a method to prevent zoonotic disease holds tremendous promise, not only to prevent animal infections, but also to protect human health, food security, and wildlife populations. The re-examination of current programs and creating new, re-energized, collaborative approaches for vaccine-preventable zoonotic diseases is needed among vaccine researchers, pharmaceutical companies, and public health and animal health professionals and producers. A One Health approach that reflects the shared burden of disease and promotes domestic and global collaborations between the human and animal health sectors and other relevant partners should be applied in strategies to prevent zoonotic diseases, including research and vaccine development, planning, funding opportunities, and implementation.

## Data Availability

Not applicable.

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
