# Peer review of "Vaccine Preventable Zoonotic Diseases: Challenges and Opportunities for Public Health Progress"

_vaccines, 2022, doi:10.3390/vaccines10070993_

Round 1
Reviewer 1 Report
1. Except for the sequence number of the first part “introduction”, the sequence numbers of other parts are missing.
2. In the section “Review of Vaccine Preventable Zoonoses”, Coronaviruses are not mentioned in the first paragraph. It's a bit abrupt to write directly about coronaviruses in the paragraph below.
3. The examples of specific vaccines used for vaccine preventable zoonoses are not comprehensive enough.
4. The section “Developing, Licensing, and Regulating Vaccines” can be simplified and put under the section “introduction”.
5. Line 361-362: “The USDA considers RVFV one of the most significant Foreign Animal Diseases”, RVFV could not be considered a disease.
6. Consider adding 1-2 figures in the manuscript.
7. The paper contains some grammatical errors and sentence structure issues. Some but not all errors are listed below:
1) Line 49-50: change "these vaccines are highly cost-effective, save human lives" to "these vaccines are highly cost-effective and could save human lives"
2) Line 65: change “evolutionary changes to the pathogen” to “evolutionary changes of the pathogen”?
3) Line 104: change “in the 1940’s” to “in the 1940s”?
4) Line 119-120: change “Oral vaccination of dogs using, live-attenuated viruses” to “Oral vaccination of dogs using live-attenuated viruses”?
5) Line 135: change “in elimination of” to “in the elimination of”?
6) Line 151: change “had a direct impact on” to “has a direct impact on”?
7) Line 171: change “Brucella vaccine use” to “the use of Brucella vaccine”
8) Line 176-179: The sentence is too long.
9) Line 236: change “Vaccine trials for MERS-CoV vaccines” to “Vaccine trials for MERS-CoV”?
10) Line 240: “though” or “through”?
11) Line 359: change “remains” to “remain”
12) Line 360-361: change “followed by rapid spread” to “followed by its rapid spread”
13) Line 406: change “is not required” to “are not required”
14) Line 430: change “and well as” to “as well as”?
15) Line 435: change “country specific” to “country-specific”
16) Line 469: change “on quality of” to “on the quality of”
17) Line 498: “analysis” and “have” are unmatched.
18) Line 513: change “highlights” to “highlight”
19) Line 577: This sentence has two periods.
Author Response
- Except for the sequence number of the first part “introduction”, the sequence numbers of other parts are missing.
Thank you for your comment. The sequence number has been removed.
- In the section “Review of Vaccine Preventable Zoonoses”, Coronaviruses are not mentioned in the first paragraph. It's a bit abrupt to write directly about coronaviruses in the paragraph below.
Thank you for your comment. Reference to emerging coronaviruses have been added to the section “Review of Vaccine Preventable Zoonoses”.
- The examples of specific vaccines used for vaccine preventable zoonoses are not comprehensive enough.
Thank you for this suggestion. This publication focuses on select vaccine-preventable zoonoses and their current vaccine programs, to guide relevant discussion points. More comprehensive references for specific vaccines (those that are currently available or that are in development) for many more zoonoses are described in Table 1.
- The section “Developing, Licensing, and Regulating Vaccines” can be simplified and put under the section “introduction”.
Thank you for your comment. These edits have been made, reflected in track changes.
- Line 361-362: “The USDA considers RVFV one of the most significant Foreign Animal Diseases”, RVFV could not be considered a disease.
Thank you for this comment. The text has been edited to reflect that RVF is a significant foreign animal disease.
- Consider adding 1-2 figures in the manuscript.
Thanks for this suggestion, After reviewing options, the authors did not see any areas where a figure would help explain matters not already detailed in the text. If the reviewers have specific suggestions for a figure, we would be happy to consider further.
- The paper contains some grammatical errors and sentence structure issues. Some but not all errors are listed below:
1) Line 49-50: change "these vaccines are highly cost-effective, save human lives" to "these vaccines are highly cost-effective and could save human lives"
This has been edited. Thank you.
2) Line 65: change “evolutionary changes to the pathogen” to “evolutionary changes of the pathogen”?
This edit has been made.
3) Line 104: change “in the 1940’s” to “in the 1940s”?
This has been edited.
4) Line 119-120: change “Oral vaccination of dogs using, live-attenuated viruses” to “Oral vaccination of dogs using live-attenuated viruses”?
This edit has been made.
5) Line 135: change “in elimination of” to “in the elimination of”?
This has been edited.
6) Line 151: change “had a direct impact on” to “has a direct impact on”?
This edit has been made.
7) Line 171: change “Brucella vaccine use” to “the use of Brucella vaccine”
This edit has been made.
8) Line 176-179: The sentence is too long.
Edited, thank you.
9) Line 236: change “Vaccine trials for MERS-CoV vaccines” to “Vaccine trials for MERS-CoV”?
This edit has been made.
10) Line 240: “though” or “through”?
This edit has been made.
11) Line 359: change “remains” to “remain”
Edited, thank you.
12) Line 360-361: change “followed by rapid spread” to “followed by its rapid spread”
Edited.
13) Line 406: change “is not required” to “are not required”
Edited.
14) Line 430: change “and well as” to “as well as”?
Edited, thank you.
15) Line 435: change “country specific” to “country-specific”
This edit has been made.
16) Line 469: change “on quality of” to “on the quality of”
Edited, thank you.
17) Line 498: “analysis” and “have” are unmatched.
Edited to match, thank you.
18) Line 513: change “highlights” to “highlight”
This edit has been made.
19) Line 577: This sentence has two periods.
The extra period has been removed.
Reviewer 2 Report
The review provides an apparent summary of U.S. efforts to prevent zoonotic diseases from a public health perspective, a pharmaceutical regulatory perspective, and a market-based perspective.
This is a common challenge not only in the U.S. but also in other developed countries and should be addressed by the health authorities in each country.
On the other hand, it is also necessary to consider interventions in hotbeds of zoonosis areas, and the paper would be more valuable if this point were also described.
As for the individual issues.
(1) What were the criteria used to select the list of pathogens in Table 1; is the list proposed by the One Health Zoonotic Disease Prioritization Workshop sufficient?
(2) Are the eight zoonotic diseases listed in L88 #1,3,4,5,6,7,9,11 in Appendix C of Reference 22 correct? It will be easier to understand if specific names are given.
(3) Concerning (2), it is noted that vaccines exist for 6 of the eight pathogens, but it would be preferable to provide specific names for these.
Author Response
The review provides an apparent summary of U.S. efforts to prevent zoonotic diseases from a public health perspective, a pharmaceutical regulatory perspective, and a market-based perspective.
This is a common challenge not only in the U.S. but also in other developed countries and should be addressed by the health authorities in each country.
On the other hand, it is also necessary to consider interventions in hotbeds of zoonosis areas, and the paper would be more valuable if this point were also described.
As for the individual issues.
(1) What were the criteria used to select the list of pathogens in Table 1; is the list proposed by the One Health Zoonotic Disease Prioritization Workshop sufficient?
The pathogens listed in Table 1 include those commonly prioritized in the One Health Zoonotic Disease Prioritization workshops, as well as those globally that present an opportunity for being vaccine preventable. The table is intended to highlight those that have vaccines, as well as the large number of zoonotic pathogens that lack vaccines to prevent them and the burden of disease that they cause.
(2) Are the eight zoonotic diseases listed in L88 #1,3,4,5,6,7,9,11 in Appendix C of Reference 22 correct? It will be easier to understand if specific names are given.
Correct, the referenced zoonotic diseases are listed in Appendix C of Reference 22. The text has been edited to reflect that six of the top eight prioritized diseases are vaccine preventable. Thank you for noting this point of confusion!
(3) Concerning (2), it is noted that vaccines exist for 6 of the eight pathogens, but it would be preferable to provide specific names for these.
Thank you for your comment. The top eight zoonotic pathogens have been listed out, and those specific vaccines that are available are included in Table 1.
Round 2
Reviewer 1 Report
The authors have addressed all my questions and comments.